# Survival machine learning methods for mortality prediction after heart transplantation in the contemporary era

**Lathan Liou**[1,2], **Elizabeth Mostofsky**[1], **Laura Lehman**[1,3,4], **Soziema Salia**[1,5], **Francisco J. Barrera**[1], **Ying Wei**[1], **Amal Cheema**[1,6], **Anuradha Lala**[7], **Andrew Beam**[1☯], **Murray A. Mittleman**[1,3,8☯]*

1 Department of Epidemiology, Harvard T.H. Chan School of Public Health, Boston, Massachusetts, United States of America, 2 Department of Genetics and Genomics Sciences, Icahn School of Medicine at Mount Sinai, New York, New York, United States of America, 3 Harvard Medical School, Boston, Massachusetts, United States of America, 4 Department of Neurology, Boston Children's Hospital, Boston, Massachusetts, United States of America, 5 Department of Internal Medicine, Cape Coast Teaching Hospital, Cape Coast, Ghana, 6 Geisel School of Medicine, Dartmouth College, Hanover, New Hampshire, United States of America, 7 Zena and Michael A. Wiener Cardiovascular Institute and Department of Population Health Science and Policy, Mount Sinai, New York, New York, United States of America, 8 Department of Medicine, Division of Cardiovascular Medicine, Beth Israel Deaconess Medical Center, Boston, Massachusetts, United States of America

☯ These authors contributed equally to this work.
* mmittlem@hsph.harvard.edu

**Data Availability Statement:** The SRTR database is based on data from transplant programs, organ procurement organizations, and histocompatibility laboratories, collected by the Organ Procurement

## Abstract

Although prediction models for heart transplantation outcomes have been developed previously, a comprehensive benchmarking of survival machine learning methods for mortality prognosis in the most contemporary era of heart transplants following the 2018 donor heart allocation policy change is warranted. This study assessed seven statistical and machine learning algorithms–Lasso, Ridge, Elastic Net, Cox Gradient Boost, Extreme Gradient Boost Linear, Extreme Gradient Boost Tree, and Random Survival Forests in a post-policy cohort of 7,160 adult heart-only transplant recipients in the Scientific Registry of Transplant Recipients (SRTR) database who received their first transplant on or after October 18, 2018. A cross-validation framework was designed in *mlr*. Model performance was also compared in a seasonally-matched pre-policy cohort. In the post-policy cohort, Random Survival Forests and Cox Gradient Boost had the highest performances with C-indices of 0.628 and 0.627. The relative importance of some predictive variables differed between the pre-policy and post-policy cohorts, such as the absence of ECMO in the post-policy cohort. Survival machine learning models provide reasonable prediction of 1-year posttransplant mortality outcomes and continual updating of prediction models is warranted in the contemporary era.

## Introduction

Orthotopic heart transplantation (HTx) remains the gold standard therapy for end-stage heart failure. Careful patient selection has allowed for a contemporary median survival of 12.5 years

and Transplantation Network. The authors had permission to use the SRTR dataset under project number #9989. The data are available from SRTR upon request by emailing srtr@srtr.org (More information can be found at https://www.srtr.org/requesting-srtr-data/data-requests/). SRTR is a third-party organization that owns the data, thus the authors are not allowed to share data per the Data Use Agreement that was signed with SRTR. The authors confirm they did not have any special privileges that other authors would not have.

**Funding:** The author(s) received no specific funding for this work.

**Competing interests:** The authors have declared that no competing interests exist.

**Abbreviations:** BMI, body mass index; CPRA, calculated panel reactive antibody; ECMO, extracorporeal membrane oxygenation; GFR, glomerular filtration rate; IABP, intra-aortic blood pump; ICU, intensive care unit; LOS, length of stay; OPTN, Organ Procurement and Transplantation Network; PCW, pulmonary capillary wedge pressure; PVR, pulmonary vascular resistance; SRTR, Scientific Registry of Transplant Recipients; VAD, ventricular assist device.

with an additional 2 years conditional upon 1-year survival post-transplant [1]. Accurate determination of which patient characteristics should be considered contraindications to transplantation for optimal outcomes remains one of the greatest challenges in our field. Historically, heart transplant risk prediction models have been generated using multivariable logistic regression [2] or Cox proportional hazard regression [3,4] and have yielded the identification of certain key risk factors. The limitation of such models lies in the assumption of purely linear relationships that do not account for more complex associations. Although many factors associated with post-transplantation survival have been identified [5,6], there remains variability in models exploring more complex relationships. Recently, several studies have attempted to improve the prediction of heart transplant outcomes using machine learning (ML) methods [7–11] and survival ML methods [12–14] in both adult and pediatric populations. Compared to traditional regression-based modeling approaches, ML algorithms can capture more complex interactions between variables. Accounting for time-to-event information tends to result in more statistically powerful prediction estimates [15]. Furthermore, posttransplantation survival is known to be non-proportional across different strata, making conventional statistical methods such as Cox proportional hazards regression potentially invalid and less powerful for assessing post-transplantation mortality among heart transplant recipients.

Another potentially important consideration is the revision of the donor heart allocation system on October 18, 2018, to "reduce wait-list mortality, enhance geographic organ sharing, and improve organ distribution equity" [16]. Although it has been shown that 1-year posttransplant survival under the new heart allocation policy was not significantly different from before [17], it is possible that both the clinical profile of transplant recipients and clinical care practices have shifted in the almost 5 years since policy implementation [18]. Thus, in this study, we explore and benchmark the performance of machine learning survival algorithms in the post-2018 transplant era (henceforth referred to as "post-policy") to better understand prognostic factors associated with one-year mortality. We compare these results to a parallel analysis run in a seasonally-matched pre-2018 transplant era cohort (henceforth referred to as "pre-policy").

## Methods

### Data source

We used data from the Scientific Registry of Transplant Recipients (SRTR). The SRTR data system includes data on all donor, waitlisted candidates, and transplant recipients in the US, submitted by the Organ Procurement and Transplantation Network (OPTN) members. The data is anonymized such that it is not possible to identify individual participants. The Health Resources and Services Administration (HRSA), U.S. Department of Health and Human Services, provides oversight of the activities of the OPTN and SRTR contractors. We had approval from SRTR under DUA #9899 to conduct this research study. Informed consent is obtained by OPTN for living donors. As formally declared by the UNOS Ethics Committee, no organs were procured from prisoners. We confirm that all research was performed in accordance with relevant guidelines/regulations. The data was last accessed on May 5, 2023. This study was reviewed and informed consent was waived by the Institutional Review Board at Harvard T.H. Chan School of Public Health due to the retrospective and anonymized nature of this registry dataset.

### Study population

We excluded recipients <18 years of age at transplant (n = 1,275) and an additional 8,187 recipients of multi-organ transplants because allocation criteria are different for multi-organ

and pediatric candidates. This study includes 7,160 adult heart-only transplant recipients who received their first transplant on or after October 18, 2018, with at least one recorded follow-up visit. We had follow-up data up until June 3, 2021. This study was reviewed and approved by the Institutional Review Board at the Harvard T.H. Chan School of Public Health.

## Data preparation

The outcome of interest was one-year all-cause mortality assessed from the time of heart transplantation to death or end of the one-year follow-up. Survival data for recipients in both cohorts were administratively censored at 1 year after transplant to prevent bias from the differential length of follow-up between cohorts. For each cohort, we split 90% as a training set and 10% as a holdout set. In order of operation, we performed z-score standardization on continuous variables, multiple imputation, and one-hot encoding on categorical variables separately in the validation set and holdout set to minimize data leakage. We included predictor variables based on expert opinion and literature review. We excluded potential predictors with over 20% missingness. For those with less than 20% missingness, we used multiple imputation with chained equations with 5 iterations using the R package *mice* [19], which preserves the associations in the data and the uncertainty in those associations. We assumed that the data are missing at random and that censoring is non-informative. We used one-hot encoding to convert categorical variables into binary variables for each level of each categorical feature. Certain variables that had many levels with only a small number of observations were combined (Supplementary Appendix). This data pre-processing led to a final set of 75 demographic, clinical, recipient, waitlist, donor, and procedural variables (114 after one-hot encoding), of which 9 were categorical, 44 binary, and the remaining 22 numeric (S1 Table in S1 File).

We also prepared a seasonally matched pre-policy cohort of patients transplants from November 1, 2014, with follow-up time to June 3, 2017, as a sensitivity analysis to account for known seasonal trends in decreased donor heart donation [20]. This cohort ends 1 year before policy implementation to avoid bias from any anticipatory practice changes before the policy implementation in October 2018.

## Machine learning analysis

We compared a Cox proportional hazards model and 7 machine learning algorithms. Lasso, Elastic Net, and Ridge regression were selected as the penalized Cox regression methods [21,22]. Boosted Cox regression methods included a Cox model with gradient boosting (Cox Boost) [23], Extreme Gradient Boosting (XGBoost) [24] with linear model-based boosting (XGBL), and tree-based boosting XGBoost (XGBT) [25]. Random Survival Forests (RSF) [26] was chosen as the tree-based ensemble method. Details on the models and selected parameters are described in the Supplementary Appendix. Each model was trained using a nested 5-repeat, 5-fold cross-validation within the validation cohort (80/20 split), and tuning was automatically performed for the machine learning algorithms to select optimal hyperparameters. A random search with 25 iterations was used to select values for the hyperparameters in the inner loop and model performance was evaluated in the outer loop. For each method, we took the parameters from the best-performing model within the cross-validation and used the entirety of the training cohort to produce the final model. The final model was then tested on the holdout set. The R package *mlr* (Machine Learning in R) [27] was used as a framework to carry out benchmarking of these experiments. A graphical schematic of our benchmark framework is presented in Fig 1. To evaluate model performance, we calculated both the cross-validated and holdout concordance index (Harrell's C-index). The C-index measures the

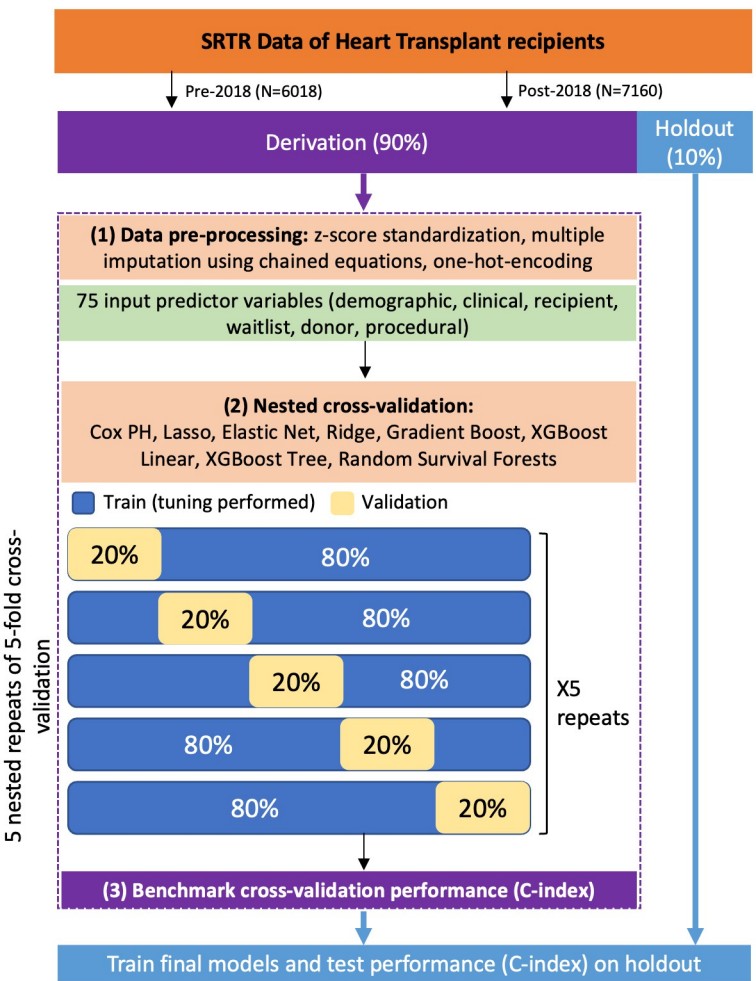

**Fig 1. Schematic of machine learning benchmarking study design.**

proportion of pairs where the observation with the higher survival time has the higher probability of survival as predicted by the model. A C-index of 1 means the predictions were perfect: higher-risk patients are ranked ahead of lower-risk patients. A C-index of 0.5 means the predictions are random: half the time higher-risk patients are correctly ranked ahead, half the time lower-risk patients are incorrectly ranked ahead. Variable weights were computed as the beta coefficients from regression-based models and importance from boosted and ensemble models. Variable relevance was determined as the number of times it was selected across 25 cross-validation iterations. Variable significance was defined as either the empirical cross-validated 95% confidence interval of the hazard ratio not crossing 1 or the importance weight not crossing 0. To assess whether there were statistical differences between cross-validated models, we calculated the corrected resampled paired t-test, proposed by Nadeau and Bengio [28], using the R package *correctR*. A standard t-test on results from a repeated k-fold cross-validation inflates the Type I error is inflated because the violation of the independence assumption leads to an underestimation in the variance. A Bonferroni adjustment was applied to p-values to control the false discovery rate at 0.05. All analyses were performed using R 4.3.0. The code used in the analysis can be found at https://github.com/latlio/srtr_mlr.

## Results

### Cohort characteristics

In the post-policy cohort, 506 patients died within one year of their heart transplant during the study period from October 18, 2018, to June 3, 2021. The crude one-year survival was 90.9% (95% CI: 90.2%, 91.7%) with 5864 total person-years of follow-up time. The average recipient age was 53.5, and the average donor age was 32.5. A total of 72.5% (n = 5193) of the cohort was male, and 72.1% (n = 5162) of the cohort was white. 27.1% had diabetes. 37.6% of patients were bridged with ventricular assist device (VAD), 4.9% of patients were bridged with extra-corporeal membrane oxygenation (ECMO), and 26.8% of patients were bridged with intra-aortic balloon pump therapy (IABP). Patients spent a median of 33 days on the wait-list and received organs that spent 3.4 hours of ischemia time on average during transport. A comprehensive summary of demographic, clinical, recipient, waitlist, donor, and procedural variables for the post-policy cohort is detailed in Table 1. We note that in the pre-policy cohort, 512 patients died within one year, and the crude one-year survival rate was 91.5% (95% CI: 90.8%, 92.2%).

**Table 1.  Baseline characteristics of heart transplant recipients from October 18, 2018 to June 3, 2021.**

|  | Overall (N = 7160) |
|---|---|
| **Allocation Status** |  |
| Previous 3-tier system | 27 (0.4%) |
| 2018 Status 1 | 587 (8.2%) |
| 2018 Status 2 | 3335 (46.6%) |
| 2018 Status 3 | 1396 (19.5%) |
| 2018 Status 4 | 1427 (19.9%) |
| 2018 Status 5 | 0 (0%) |
| 2018 Status 6 | 388 (5.4%) |
| **BMI (kg/m2)** |  |
| Mean (SD) | 27.8 (5.03) |
| Median [Min, Max] | 27.5 [11.7, 47.5] |
| **Highest Education Received** |  |
| Below High School | 243 (3.4%) |
| High School Graduate | 2549 (35.6%) |
| College Graduate and beyond | 4036 (56.4%) |
| **Gender** |  |
| Male | 5193 (72.5%) |
| Female | 1967 (27.5%) |
| **Insurance** |  |
| Medicaid | 968 (13.5%) |
| Medicare | 2372 (33.1%) |
| Private | 3262 (45.6%) |
| Other | 314 (4.4%) |
| **Age** |  |
| Mean (SD) | 53.5 (12.9) |
| Median [Min, Max] | 57.0 [18.0, 76.0] |
| **Cardiac Etiology** |  |
| Valvular Heart Disease | 74 (1.0%) |
| Dilated Myopathy | 1302 (18.2%) |
| Ischemic Dilated Myopathy | 1900 (26.5%) |

*(Continued)*

**Table 1.** (Continued)

| | Overall (N = 7160) |
|---|---|
| Restrictive Myopathy | 330 (4.6%) |
| Coronary Artery Disease | 132 (1.8%) |
| Other | 653 (9.1%) |
| Idiopathic | 2701 (37.7%) |
| **Dialysis** | |
| No | 7049 (98.4%) |
| Yes | 111 (1.6%) |
| **Type 2 Diabetes** | |
| No | 5174 (72.3%) |
| Yes | 1938 (27.1%) |
| **Previous Malignancy** | |
| No | 6530 (91.2%) |
| Yes | 630 (8.8%) |
| **Prior Cardiac Surgery** | |
| No | 5644 (78.8%) |
| Yes | 1202 (16.8%) |
| **Steroid Use** | |
| No | 6441 (90.0%) |
| Yes | 353 (4.9%) |
| **Cytomegalovirus** | |
| Negative | 3075 (42.9%) |
| Positive | 3689 (51.5%) |
| **Epstein-Barr Virus** | |
| Negative | 646 (9.0%) |
| Positive | 5910 (82.5%) |
| **Functional Status At Time of Transplant** | |
| Moribund/hospitalized/severely disabled | 4302 (60.1%) |
| Significant Assistance | 1638 (22.9%) |
| Normal | 518 (7.2%) |
| **HBV Antibody** | |
| Negative | 6325 (88.3%) |
| Positive | 307 (4.3%) |
| **HBV Surf Antigen** | |
| Negative | 6664 (93.1%) |
| Positive | 100 (1.4%) |
| **Hepatitis C Virus** | |
| Negative | 6593 (92.1%) |
| Positive | 149 (2.1%) |
| **HIV Status** | |
| No | 6733 (94.0%) |
| Yes | 37 (0.5%) |
| **Time on Waitlist** | |
| Mean (SD) | 186 (385) |
| Median [Min, Max] | 33.0 [0, 5150] |
| **Blood Type** | |
| A | 2852 (39.8%) |
| AB | 370 (5.2%) |

(*Continued*)

**Table 1.** (Continued)

| | Overall (N = 7160) |
|---|---|
| B | 1100 (15.4%) |
| O | 2838 (39.6%) |
| **Accept a Donor After Cardiac Death** | |
| No | 6720 (93.9%) |
| Yes | 440 (6.1%) |
| **Accept a Hepatitis B Positive Donor** | |
| No | 3167 (44.2%) |
| Yes | 3993 (55.8%) |
| **Accept a Hepatitis C Positive Donor** | |
| No | 2401 (33.5%) |
| Yes | 4759 (66.5%) |
| **Accept a Donor with History of Heart Disease** | |
| No | 2856 (39.9%) |
| Yes | 3218 (44.9%) |
| **Symptomatic Cerebrovascular Disease** | |
| No | 6556 (91.6%) |
| Yes | 512 (7.2%) |
| **Candidate Functional Status** | |
| Moribund/hospitalized/severely disabled | 3870 (54.1%) |
| Significant Assistance | 2275 (31.8%) |
| Normal | 663 (9.3%) |
| **Candidate Preliminary Cross-Matching Required** | |
| No | 6642 (92.8%) |
| Yes | 518 (7.2%) |
| **Waitlist Glomerular Filtration Rate (MDRD)** | |
| Mean (SD) | 68.7 (27.8) |
| Median [Min, Max] | 64.2 [4.91, 522] |
| **Glomerular Filtration Rate (MDRD) At Time of Transplant** | |
| Mean (SD) | 70.1 (30.0) |
| Median [Min, Max] | 65.1 [6.84, 494] |
| **Number of HLA-A Mismatches** | |
| Mean (SD) | 1.51 (0.603) |
| Median [Min, Max] | 2.00 [0, 2.00] |
| **Number of HLA-B Mismatches** | |
| Mean (SD) | 1.73 (0.481) |
| Median [Min, Max] | 2.00 [0, 2.00] |
| **Number of HLA-DR Mismatches** | |
| Mean (SD) | 1.56 (0.580) |
| Median [Min, Max] | 2.00 [0, 2.00] |
| **Received ECMO Pre-transplant** | |
| No | 6810 (95.1%) |
| Yes | 350 (4.9%) |
| **Specific HLA Antibody** | |
| No | 5674 (79.2%) |
| Yes | 728 (10.2%) |
| **HLA Typing Done** | |
| No | 525 (7.3%) |

(*Continued*)

**Table 1.** (Continued)

| | Overall (N = 7160) |
|---|---|
| Yes | 6411 (89.5%) |
| **Received IABP** | |
| No | 5238 (73.2%) |
| Yes | 1922 (26.8%) |
| **Received Inotropes** | |
| No | 4531 (63.3%) |
| Yes | 2629 (36.7%) |
| **Received Life Support (e.g. ECMO, IABP, PGE, Inotropes, Ventilator, NO, VAD)** | |
| No | 1196 (16.7%) |
| Yes | 5663 (79.1%) |
| **Received Other Type of Life Support** | |
| No | 6791 (94.8%) |
| Yes | 369 (5.2%) |
| **Medical Condition** | |
| ICU | 3559 (49.7%) |
| Hospitalized Not in ICU | 907 (12.7%) |
| Not Hospitalized | 2389 (33.4%) |
| **Number of HLA Mismatches** | |
| Mean (SD) | 4.69 (1.01) |
| Median [Min, Max] | 5.00 [0, 6.00] |
| **Total Bilirubin (mg/dL)** | |
| Mean (SD) | 1.01 (1.78) |
| Median [Min, Max] | 0.700 [0.100, 51.0] |
| **Received Transfusion(s)** | |
| No | 5861 (81.9%) |
| Yes | 950 (13.3%) |
| **Received Ventilator Support** | |
| No | 5758 (80.4%) |
| Yes | 1061 (14.8%) |
| **VAD Usage** | |
| No | 4232 (59.1%) |
| Yes | 2695 (37.6%) |
| **Donor Blood Type** | |
| A | 2557 (35.7%) |
| AB | 134 (1.9%) |
| B | 787 (11.0%) |
| O | 3682 (51.4%) |
| **Donor Age** | |
| Mean (SD) | 32.5 (10.6) |
| Median [Min, Max] | 32.0 [9.00, 70.0] |
| **Donor Anti-CMV Antibody** | |
| Negative | 2735 (38.2%) |
| Positive | 4392 (61.3%) |
| **Donor Anti-HCV Antibody** | |
| Negative | 6345 (88.6%) |
| Positive | 815 (11.4%) |
| **Donor Cause of Death** | |

(*Continued*)

**Table 1.** (Continued)

| | Overall (N = 7160) |
|---|---|
| Anoxia | 3202 (44.7%) |
| Cerebrovascular/Stroke | 956 (13.4%) |
| Head Trauma | 2795 (39.0%) |
| CNS Tumor | 23 (0.3%) |
| **Donor Serum Creatinine (mg/dL)** | |
| Mean (SD) | 1.67 (1.76) |
| Median [Min, Max] | 1.04 [0.0400, 17.9] |
| **DDVAP Administered to Donor** | |
| No | 6433 (89.8%) |
| Yes | 552 (7.7%) |
| **Donor Gender** | |
| Male | 5132 (71.7%) |
| Female | 2028 (28.3%) |
| **Donor Height (cm)** | |
| Mean (SD) | 174 (9.45) |
| Median [Min, Max] | 175 [122, 206] |
| **Donor Serum Creatinine >1.5 mg/dL** | |
| No | 5164 (72.1%) |
| Yes | 1994 (27.8%) |
| **Donor History of Cancer** | |
| No | 7008 (97.9%) |
| Yes | 66 (0.9%) |
| **Donor Cigarette Use >20 Pack-years** | |
| No | 6149 (85.9%) |
| Yes | 855 (11.9%) |
| **Donor History of Cocaine Use** | |
| No | 4981 (69.6%) |
| Yes | 1875 (26.2%) |
| **Donor History of Diabetes** | |
| No | 278 (3.9%) |
| Yes | 6816 (95.2%) |
| **Donor History of Hypertension** | |
| No | 1104 (15.4%) |
| Yes | 5982 (83.5%) |
| **Donor History of Drug Use** | |
| No | 2530 (35.3%) |
| Yes | 4372 (61.1%) |
| **Donor History of Inotrope Use** | |
| No | 4530 (63.3%) |
| Yes | 2448 (34.2%) |
| **Donor Cardiac Arrest** | |
| No | 6487 (90.6%) |
| Yes | 673 (9.4%) |
| **Donor Weight (kg)** | |
| Mean (SD) | 84.9 (20.2) |
| Median [Min, Max] | 81.8 [39.9, 225] |
| **Recipient Cardiac Output (L/min)** | |

(*Continued*)

**Table 1.** (Continued)

| | Overall (N = 7160) |
|---|---|
| Mean (SD) | 4.43 (1.43) |
| Median [Min, Max] | 4.27 [0.200, 14.3] |
| **Total Ischemic Time (min)** | |
| Mean (SD) | 206 (64.8) |
| Median [Min, Max] | 205 [20.0, 720] |
| **Pulmonary Capillary Wedge Pressure (mmHg)** | |
| Mean (SD) | 17.8 (8.73) |
| Median [Min, Max] | 17.0 [0, 50.0] |
| **Transplant Procedure Type** | |
| Orthotopic Bicaval | 5677 (79.3%) |
| Orthotopic Traditional | 977 (13.6%) |
| Orthotopic Total | 188 (2.6%) |
| Heterotopic | 0 (0%) |
| **Arterial Diastolic Blood Pressure (mmHg)** | |
| Mean (SD) | 19.3 (8.67) |
| Median [Min, Max] | 18.0 [0, 110] |
| **Arterial Mean Blood Pressure (mmHg)** | |
| Mean (SD) | 27.0 (10.1) |
| Median [Min, Max] | 26.0 [0, 110] |
| **Arterial Systolic Blood Pressure (mmHg)** | |
| Mean (SD) | 39.6 (13.8) |
| Median [Min, Max] | 38.0 [0, 158] |

## Machine learning benchmark performance

In the post-policy holdout data, RSF performed the best with a C-index of 0.628, while Cox performed the worst with a C-index of 0.585 (Table 2). Cox Boost performed similarly well with a C-index of 0.627. Both Lasso and Elastic Net performed moderately well in the holdout with C-indices of 0.613 each, although their performance in the cross-validation was poor with cross-validated mean C-indices of 0.516 and 0.508 respectively (S3 Table in S1 File). The average cross-validated C-index ranged from 0.615 to 0.508 (Fig 2). Although XGBL had the highest cross-validation C-index (mean = 0.615 [SD = 0.025]), it did not have the highest holdout C-index. The most variable algorithms were Lasso (SD = 0.033) and Ridge (SD = 0.031) across

**Table 2. C-indices for holdout data.**

| Model | Pre-Policy Model Tested on Pre-Policy Holdout | Post-Policy Model Tested on Post-Policy Holdout | Pre-Policy Model Tested on Post-Policy Holdout |
|---|---|---|---|
| Coxph | 0.610 | 0.585 | 0.573 |
| Ridge | 0.617 | 0.586 | 0.577 |
| Lasso | 0.626 | 0.613 | 0.616 |
| Elastic Net | 0.646 | 0.613 | 0.620 |
| Cox Boost | 0.654 | 0.627 | 0.633 |
| XGBoost Linear | 0.646 | 0.600 | 0.585 |
| XGBoost Tree | 0.650 | 0.612 | 0.631 |
| Random Survival Forests | 0.656 | 0.628 | 0.616 |

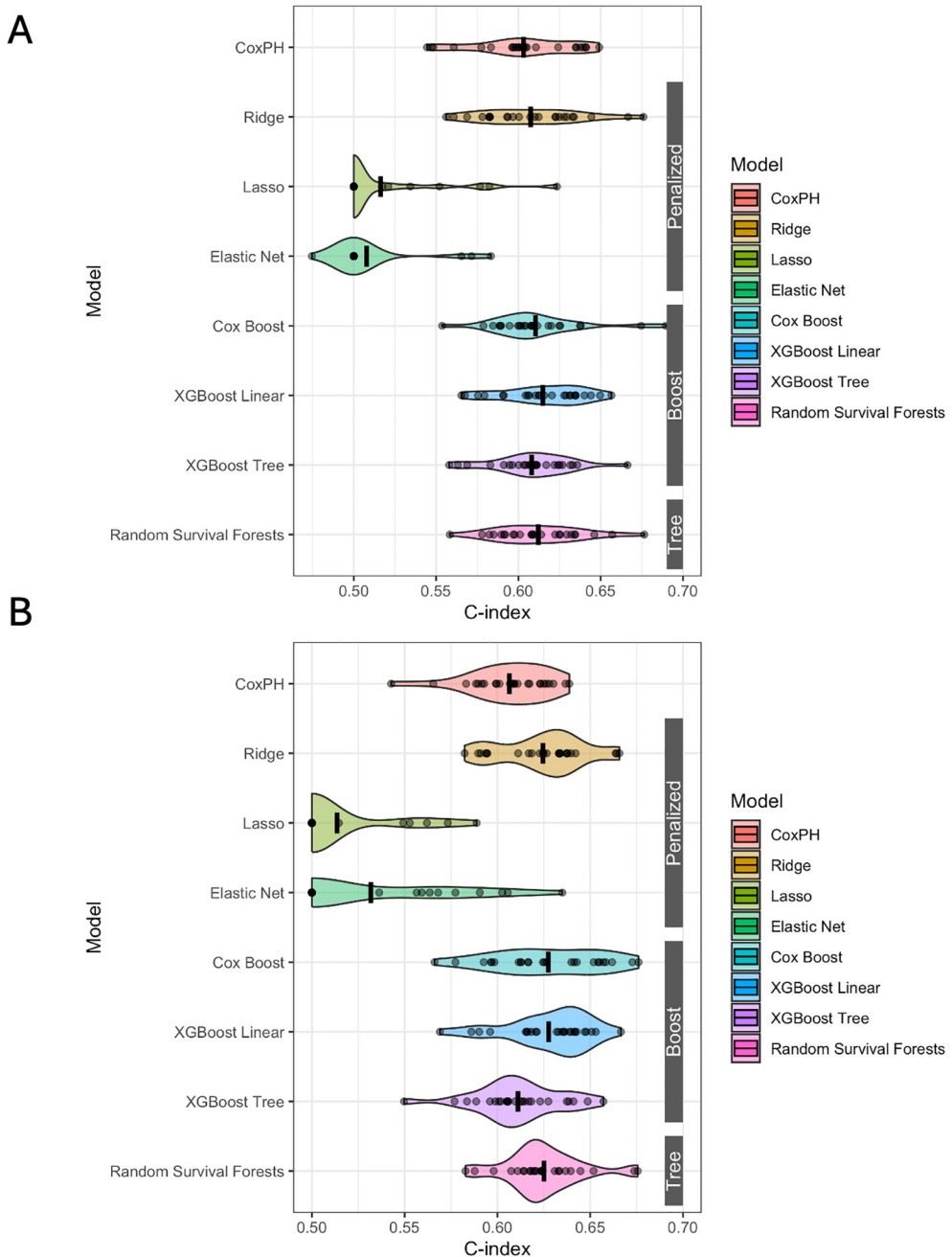

**Fig 2. Violin plots comparing cross-validation C-indices for 7 different machine learning survival models against Cox PH in the A) post-policy cohort and B) pre-policy cohort.** Black bar indicates the mean c-index.

cross-validation repeats. Except for Lasso and Elastic Net, there were no statistically significant differences in cross-validation C-indices between Cox and the ML methods (S4 Table in S1 File). Lasso and Elastic Net had significantly worse cross-validated performance than the other ML methods.

In the pre-policy cohort, the algorithms performed similarly, with Cox Boost (holdout C-index = 0.633) performing the best and Cox (holdout C-index = 0.573) the worst (Table 2).

Cross-validation C-indices ranged from 0.628 to 0.514, and the most variable algorithm was Lasso (SD = 0.044) (S4 Table in S1 File). There were also largely no statistically significant differences in cross-validation C-indices between Cox and the ML methods except for Lasso (S4 Table in S1 File). All of the pre-policy cohort-trained models applied to the post-policy holdout set performed more poorly compared to the pre-policy holdout set.

### Prognostic variables for posttransplant one-year mortality

Sparse models such as Lasso, Elastic Net, and Cox Boost use a regularization term to shrink coefficients of variables to zero, effectively generating a subset of variables. Hereafter, we refer to this as a model's "selection", although we clarify that we did not perform computational feature selection before training the model. Since we performed 5 repeats of 5-fold cross-validation, a variable can be selected 25 times at most. We use this crude measure of selection to first determine which variables tend to be included in the majority of model replications [29], although we note that there exists inherent variability in sparse variable selection [30]. Age, total ischemic time, bilirubin, BMI, and ECMO were the top 5 most selected variables in the post-policy cohort (Fig 3). The top 5 in the pre-policy cohort were similar except for HIV status. VAD, GFR, prior transfusions, and Medicaid were other variables that were frequently selected in the post- and pre-policy cohorts. Comprehensive heatmap summaries of the total number of times each input predictor was selected in each model in the post- and pre-policy cohorts are provided in S1 and S2 Figs in S1 File respectively.

Plots of the top 20 most important features for the top-performing models are shown in Fig 4. The most important predictor variables are fairly constant between post-policy and pre-policy cohorts, although there are some differences. Some of the most important variables RSF selected in the post-policy cohort were bilirubin, age, pulmonary diastolic pressure, donor height, and wait time (Fig 4A). In the pre-policy cohort, bilirubin, donor age, eGFR measured at waitlist and pre-transplant, and age were selected (Fig 4B). Cox Boost selected HCV positive as the most important variable in the post-policy cohort and HIV negative as the most important variable in the pre-policy cohort (Fig 4C and 4D). XGBT selected bilirubin, eGFR, age, and BMI as the most important variables in both pre and post-policy cohorts (Fig 4E and 4F). Boxplots for all variables hazard ratios and importance scores for all models are shown in S3-9 Figs in S1 File.

## Discussion

In this study, we benchmarked survival machine learning methods for the prognosis of one-year all-cause mortality in adult heart transplant recipients transplanted after the UNOS allocation policy change in October 2018 [16]. We also compared the models in a pre-policy era to assess whether the characteristics of predictor variables changed. In both the post- and pre-policy eras, the discriminatory power of machine learning models for one-year all-cause mortality tended to be higher compared to Cox, although not statistically different according to cross-validated C-indices. While the ML models were not statistically superior to Cox, these complex models likely have an advantage over Cox regression in two aspects: they do not require the proportional hazards assumption and they can capture nonlinear interactions between outcome and variables. Thus, these models can potentially provide a level of insight into the different predictive factors in these two eras that linear models cannot. It is also important to state that first, we only used information available pretransplant since that is where a post-transplant mortality prediction model is most likely useful, unlike other prediction models [31]. This may have limited the maximum possible performance of machine learning performance on mortality prediction. Second, based on our study design, we did not use

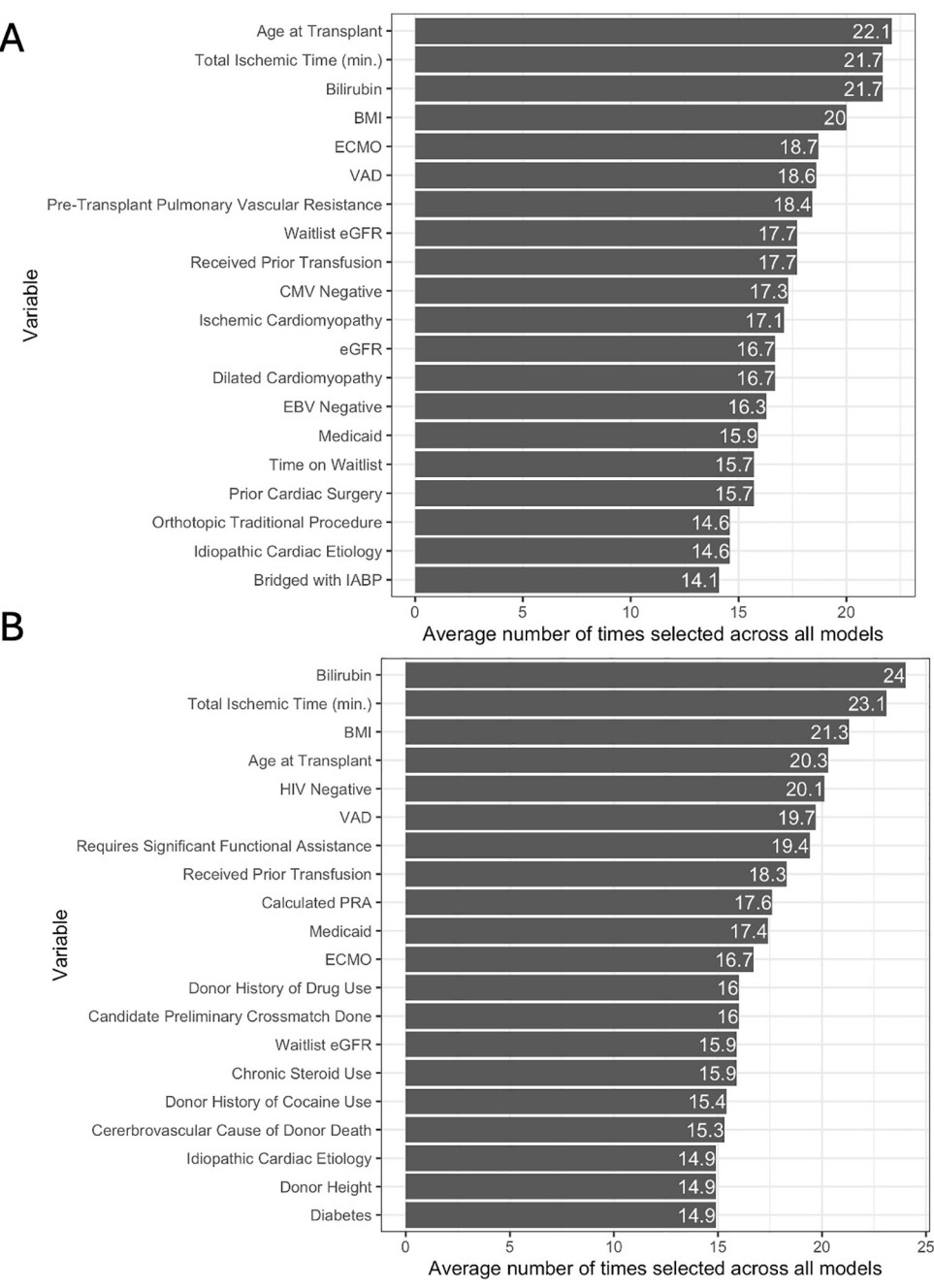

**Fig 3. Average number of times variable is selected across all sparse machine learning models in the A) post-policy cohort and B) pre-policy cohort.** Sparse machine learning models include Lasso, Elastic Net, and Cox Boost.

the full range of data available, as other predictive models have done, so our model performance may be affected by lower statistical power. The pre-policy cohort-trained models had lower C-indices when applied to a post-policy holdout set compared to a pre-policy holdout set. This could be explained by underlying changes in treatment practices following the policy change in 2018 such as increased IABP usage and increased pre-transplant temporary mechanical circulatory support use [17,18,32–34].

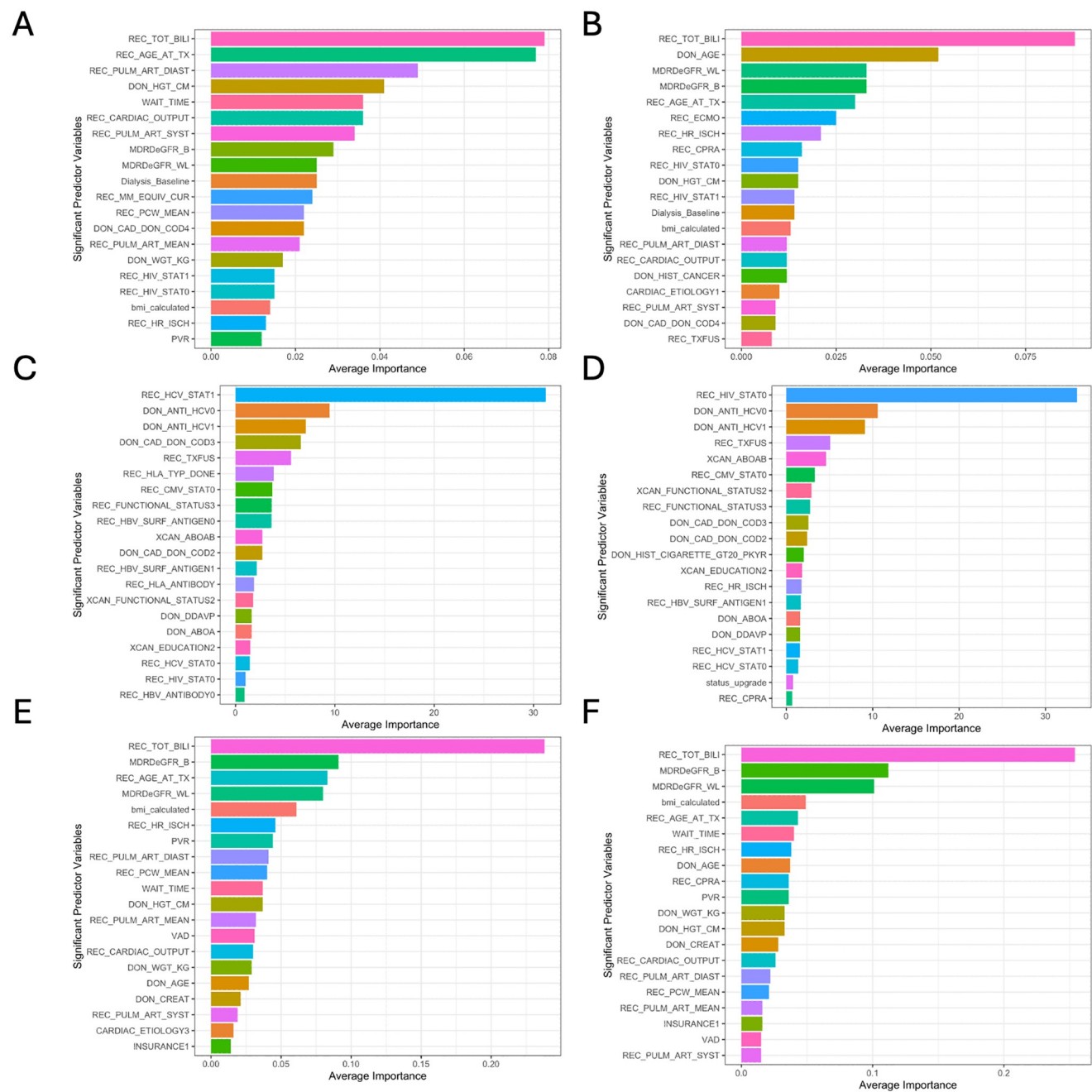

**Fig 4. Top 20 significant predictor variables.** A) RSF in post-policy cohort B) RSF in pre-policy cohort C) Cox Boost in post-policy cohort D) Cox Boost in pre-policy cohort E) XGBT in post-policy cohort F) XGBT in pre-policy cohort.

Although we begin with the caveat that the variables assessed by these survival machine learning models should not be taken as causal nor be interpreted too heavily, we briefly highlight two observations. The first is that ECMO was considered an important variable by RSF in the pre-policy cohort, but not in the post-policy cohort. Given the ongoing debate on whether ECMO is associated with worse post-transplant outcomes [32,34–39], the absence of ECMO in the contemporary post-policy model may suggest that it is no longer a highly

discriminative variable of one-year mortality relative to other variables. Whether this is due to improvements in ECMO management or increased relative usage of IABP remains a future direction of research. Additionally, HIV status was no longer considered an important variable in RSF and Cox Boost in the post-policy cohort. Although one study looked found no differences in transplantation outcomes for HIV-positive and -negative patients overall since 1987 [40], a question arises as to whether contemporary management of HIV patients has improved.

## Study limitations

First, as with most national registries, SRTR is susceptible to data entry errors and missing data regarding organ recovery or failure necessary to create continuously updated mortality estimates. However, the SRTR conducts edit checks, validation of data at time of entry, and internal verification when there are outliers. Second, the data did not include known prognostic risk factors such as natriuretic peptides, serum sodium, and specific hemodynamic data such as heart rate, although these are usually available and collected at individual centers. In addition, SRTR does not include other variables such as patient adherence and granular clinical decision-making variables that likely affect survival. Notably, status justification form variables had a high proportion of missingness (above 90%), so they were not included in the predictor set. We would also like to mention that we did not have data that adequately capture social determinants of health relating to access to care or socioeconomic status. Third, this study is exploratory, and the intent was not to build a prediction model for individual prediction of mortality risk but rather to identify potentially important variables associated with mortality. Fourth, machine learning models are often more difficult to interpret than linear models such as Cox since many hyperparameters can be tuned and there is no easily interpretable measure of directional association. Fifth, we assumed that there was only missingness at random and non-informative censoring. However, there may be informative censoring, since early post-transplant deaths are more likely to be recorded than routine follow-up appointments in the first year after transplantation. Sixth, we did not have access to an external heart transplant dataset, which is needed to generalize our findings to populations outside of the United States. Future studies can test survival machine learning methods on prospectively collected SRTR data or more carefully assess how associations of individual predictive variables have changed following the UNOS 2018 policy change.

## Conclusions

In this benchmark prognostic study, we demonstrated that machine learning models demonstrate reasonable one-year mortality prediction and can help reveal complex relationships between predictor variables and mortality. We show differences in important variables such as ECMO and HIV status between the post-policy and pre-policy eras, which showcases the ability of machine learning models to generate future hypotheses for research and suggest the need for continual updating of existing models. As clinical care practices continue to evolve, exploring computational time-to-event algorithms to develop mortality risk prediction models with more accumulated data is warranted.

## Supporting information

**S1 File.**
(DOCX)

## Acknowledgments

We would like to thank Annette Spooner for sharing her machine learning benchmark code with us and for helping with some of the technical components of the analysis.

## Author Contributions

**Conceptualization:** Lathan Liou, Murray A. Mittleman.

**Data curation:** Elizabeth Mostofsky.

**Formal analysis:** Lathan Liou.

**Investigation:** Lathan Liou, Murray A. Mittleman.

**Methodology:** Lathan Liou, Anuradha Lala, Andrew Beam, Murray A. Mittleman.

**Project administration:** Elizabeth Mostofsky, Murray A. Mittleman.

**Resources:** Elizabeth Mostofsky, Murray A. Mittleman.

**Software:** Lathan Liou.

**Supervision:** Anuradha Lala, Andrew Beam, Murray A. Mittleman.

**Visualization:** Lathan Liou.

**Writing – original draft:** Lathan Liou.

**Writing – review & editing:** Lathan Liou, Elizabeth Mostofsky, Laura Lehman, Soziema Salia, Francisco J. Barrera, Ying Wei, Amal Cheema, Anuradha Lala, Andrew Beam, Murray A. Mittleman.

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
