## [Decision Letter · Decision Letter 0]

2 Oct 2024

PONE-D-24-16546Survival Machine Learning Methods for Mortality Prediction After Heart Transplantation in the Contemporary EraPLOS ONE

Dear Dr. Liou,

Thank you for submitting your manuscript to PLOS ONE. After careful consideration, we feel that it has merit but does not fully meet PLOS ONE’s publication criteria as it currently stands. Therefore, we invite you to submit a revised version of the manuscript that addresses the points raised during the review process.

We look forward to receiving your revised manuscript.

Kind regards,

Marsa Gholamzadeh, PhD

Academic Editor

PLOS ONE

Journal Requirements:

1. When submitting your revision, we need you to address these additional requirements. Please ensure that your manuscript meets PLOS ONE's style requirements, including those for file naming. The PLOS ONE style templates can be found at https://journals.plos.org/plosone/s/file?id=wjVg/PLOSOne_formatting_sample_main_body.pdf and https://journals.plos.org/plosone/s/file?id=ba62/PLOSOne_formatting_sample_title_authors_affiliations.pdf 2. Please note that PLOS ONE has specific guidelines on code sharing for submissions in which author-generated code underpins the findings in the manuscript. In these cases, we expect all author-generated code to be made available without restrictions upon publication of the work. Please review our guidelines at https://journals.plos.org/plosone/s/materials-and-software-sharing#loc-sharing-code and ensure that your code is shared in a way that follows best practice and facilitates reproducibility and reuse. 3. For studies involving third-party data, we encourage authors to share any data specific to their analyses that they can legally distribute. PLOS recognizes, however, that authors may be using third-party data they do not have the rights to share. When third-party data cannot be publicly shared, authors must provide all information necessary for interested researchers to apply to gain access to the data. (https://journals.plos.org/plosone/s/data-availability#loc-acceptable-data-access-restrictions)  For any third-party data that the authors cannot legally distribute, they should include the following information in their Data Availability Statement upon submission:1) A description of the data set and the third-party source2) If applicable, verification of permission to use the data set3) Confirmation of whether the authors received any special privileges in accessing the data that other researchers would not have4) All necessary contact information others would need to apply to gain access to the data 4. Please include captions for your Supporting Information files at the end of your manuscript, and update any in-text citations to match accordingly. Please see our Supporting Information guidelines for more information: http://journals.plos.org/plosone/s/supporting-information.

Reviewers' comments:

Reviewer's Responses to Questions

**Comments to the Author**

1. Is the manuscript technically sound, and do the data support the conclusions?

Reviewer #1: Yes

Reviewer #2: Yes

2. Has the statistical analysis been performed appropriately and rigorously? 

Reviewer #1: Yes

Reviewer #2: Yes

3. Have the authors made all data underlying the findings in their manuscript fully available?

Reviewer #1: Yes

Reviewer #2: Yes

4. Is the manuscript presented in an intelligible fashion and written in standard English?

Reviewer #1: Yes

Reviewer #2: Yes

5. Review Comments to the Author

Reviewer #1: Thanks for this important contribution. It might be worthwhile explaining the C-statistic in more detail - especially for non-experts. This exploration of ML in this space was first carried out by our group avout 10 years ago and did not see the light of publication. So, here's hoping you will.

Reviewer #2: The study's comprehensive analysis of survival machine learning algorithms and its potential clinical implications make it a significant addition to the literature. The authors have effectively addressed the key research questions and provided a well-structured presentation of their findings.

The manuscript is technically sound, with appropriate statistical analysis, and the data fully support the conclusions. The presentation is clear and intelligible, written in standard English. Therefore, it is recommended to be accepted for publication without the need for a review.

6. PLOS authors have the option to publish the peer review history of their article (what does this mean?). If published, this will include your full peer review and any attached files.

Reviewer #1: **Yes: **Jaishankar Raman

Reviewer #2: **Yes: **Amr H. Zyoud

---

## [Author Response · Author response to Decision Letter 0]

21 Oct 2024

PONE-D-24-16546

Survival Machine Learning Methods for Mortality Prediction After Heart Transplantation in the Contemporary Era 

Reviewer #1: Thanks for this important contribution. It might be worthwhile explaining the C-statistic in more detail - especially for non-experts. This exploration of ML in this space was first carried out by our group avout 10 years ago and did not see the light of publication. So, here's hoping you will.

• We thank Reviewer #1 for their encouraging comments. We have added in our Methods, page 8 more accessible language describing the C-statistic in more detail: “A C-index of 1 means the predictions were perfect: higher-risk patients are ranked ahead of lower-risk patients. A C-index of 0.5 means the predictions are random: half the time higher-risk patients are correctly ranked ahead, half the time lower-risk patients are incorrectly ranked ahead.”

Reviewer #2: The study's comprehensive analysis of survival machine learning algorithms and its potential clinical implications make it a significant addition to the literature. The authors have effectively addressed the key research questions and provided a well-structured presentation of their findings.

The manuscript is technically sound, with appropriate statistical analysis, and the data fully support the conclusions. The presentation is clear and intelligible, written in standard English. Therefore, it is recommended to be accepted for publication without the need for a review.

• We thank Reviewer #2 for their encouraging comments towards publication acceptance.

---

## [Editor Report · Decision Letter 1]

29 Oct 2024

Survival Machine Learning Methods for Mortality Prediction After Heart Transplantation in the Contemporary Era

PONE-D-24-16546R1

Dear Dr. Liou,

We’re pleased to inform you that your manuscript has been judged scientifically suitable for publication and will be formally accepted for publication once it meets all outstanding technical requirements.

Kind regards,

Marsa Gholamzadeh, PhD

Academic Editor

PLOS ONE
---

## [Editor Report · Acceptance letter]

7 Nov 2024

PONE-D-24-16546R1 

PLOS ONE

Dear Dr. Liou, 

I'm pleased to inform you that your manuscript has been deemed suitable for publication in PLOS ONE. Congratulations! Your manuscript is now being handed over to our production team.

Kind regards, 

on behalf of

Dr. Marsa Gholamzadeh 

Academic Editor

PLOS ONE